# Environmental Factors in Northern Italy and Sickle Cell Disease Acute Complications: A Multicentric Study

**DOI:** 10.3390/children9101478

**Published:** 2022-09-27

**Authors:** Maria Elena Guerzoni, Stefano Marchesi, Giovanni Palazzi, Mariachiara Lodi, Margherita Pinelli, Donatella Venturelli, Elena Bigi, Nadia Quaglia, Paola Corti, Roberta Serra, Raffaella Colombatti, Laura Sainati, Nicoletta Masera, Francesca Colombo, Angelica Barone, Lorenzo Iughetti

**Affiliations:** 1Pediatric Unit, Department of Medical and Surgical Sciences of Mothers, Children and Adults, University of Modena and Reggio Emilia, 41121 Modena, Italy; 2Arpae-Agenzia Regionale per la Prevenzione, L’ambiente e L’energia dell’Emilia, Romagna, 41121 Modena, Italy; 3Oncology and Hematology Pediatric Unit, Department of Medical and Surgical Sciences for Mothers, Children and Adults, University of Modena and Reggio Emilia, 41121 Modena, Italy; 4Department of Medical and Surgical Sciences for Mothers, Child and Adults, Post Graduate School of Pediatrics, University of Modena and Reggio Emilia, 41121 Modena, Italy; 5Department of Transfusion Medicine, University Hospital of Modena, 41124 Modena, Italy; 6Clinic of Pediatric Hematology Oncology, Department of Woman’s and Child’s Health, University Hospital of Padova, 35128 Padova, Italy; 7Department of Pediatrics, University of Milano Bicocca, Fondazione MBBM, 20900 Monza, Italy; 8Pediatric Onco-Hematology, Department of Pediatrics, University Hospital of Parma, 43126 Parma, Italy

**Keywords:** sickle cell disease, environmental factors, air quality, meteorological parameters, acute chest syndrome, vase-occlusive crises

## Abstract

Background: Environmental factors seem to influence clinical manifestations of sickle cell disease (SCD), but few studies have shown consistent findings. We conducted a retrospective multicentric observational study to investigate the influence of environmental parameters on hospitalization for vaso-occlusive crises (VOC) or acute chest syndrome (ACS) in children with SCD. Methods: Hospital admissions were correlated with daily meteorological and air-quality data obtained from Environmental Regional Agencies in the period 2011–2015. The effect of different parameters was assessed on the day preceding the crisis up to ten days before. Statistical analysis was performed using a quasi-likelihood Poisson regression in a generalized linear model. Results: The risk of hospitalization was increased for low maximum temperature, low minimum relative humidity, and low atmospheric pressure and weakly for mean wind speed. The diurnal temperature range and temperature difference between two consecutive days were determined to be important causes of hospitalization. For air quality parameters, we found a correlation only for high levels of ozone and for low values at the tail corresponding to the lowest concentration of this pollutant. Conclusions: Temperature, atmospheric pressure, humidity and ozone levels influence acute complications of SCD. Patients’ education and the knowledge of the modes of actions of these factors could reduce hospitalizations.

## 1. Introduction

SCD is the most common genetic disease in the world, with a prevalence among newborns ranging from 0.1/1000 in non-endemic countries to 20/1000 in several parts of Africa. In the last 15 years in Italy, the disease prevalence has changed due to immigration fluxes from Africa, Balkans and South America mainly to North Italy regions, with an estimation of almost 1000–1500 children and 2500–3000 patients overall with SCD living in Italy [1]. SCD is characterized by a significant clinical heterogeneity, some patients remain almost asymptomatic, whereas others present frequent complications. Scientific progress in the study of the physiopathological mechanisms of this hematological disorder contributed to the development of new pharmacological perspectives [2].

Genetic factors only partially explain this variability, whereas environmental factors (climate and air quality) seem to have a relevant role [3]. This is already recognized in clinical practice: patients are warned to avoid low oxygen tensions, extreme temperatures and dehydration [4,5,6]. Furthermore, studies demonstrated the importance of infective trigger and seasonality for VOC and for ACS in Italy [7,8]. Nevertheless, few studies investigated the influence of environment on SCD. Recent studies considered samples of adult and pediatric patients and a long period of follow-up; nevertheless, different findings emerged [9,10,11,12,13]. Daily levels of pollutants have important effects on health; the impact of any single factor is difficult to evaluate because almost all pollutants are related to each other. Particulate matter, nitrogen dioxide (NO_2_) and ozone (O_3_) levels are Europe’s most serious pollutants for human health, especially for children, older people and those affected by pre-existing illnesses [14]. Patients with SCD presenting frequent hospitalizations have difficulty in attending school and work, limiting social relations and thus promoting a worsening of quality of life [15]. Identification of the role of environment could promote preventive interventions to relieve the burden of this disease. The aim of the study is to investigate the presence of a correlation between environmental factors and the number of emergency department accesses for VOC and ACS in pediatric and young adult patients with SCD living in Northern Italy, where the larger Sickle Cell Centers are located [16].

## 2. Materials and Methods

### 2.1. Patients

The study was retrospectively conducted using data collected during a 5-year period (from 1 January 2011 to 31 December 2015, 1826 days). We considered Emergency Department (ED) visits for VOC and ACS of all patients followed for SCD in four Pediatric Onco-Hematology Centers located in the cities of Modena, Parma, Monza and Padova, all of them located in the Po Valley in Northern Italy. The inclusion criteria were a diagnosis of SCD with S/S, S/β thalassemia or S/C genotype and a VOC and/or ACS diagnosis. We also included young adult patients still followed-up from the pediatric center. VOC was defined as a new onset pain that needs analgesics use, with exclusion of trauma. ACS was defined as an acute illness characterized by the finding of a new pulmonary infiltrate consistent with consolidation with one or more respiratory symptoms or signs (cough, chest pain, fever, hypoxemia and tachypnea). Clinical data were retrospectively collected from clinical reports. Due to the small sample size and considering the geographical proximity, Parma data have been merged with Modena data. The study was approved by local ethics committee.

### 2.2. Environmental Parameters

Several environmental variables have been collected from Environmental Regional Agencies, also according to previous studies in the literature. Almost all of them are routinely measured in meteorological and air quality stations. The meteorological variables are the atmospheric pressure, temperature, relative humidity and wind velocity. Minimum, mean and maximum values for each variable have been considered in the analysis separately. Atmospheric pressure is directly related to good and bad weather conditions. Wind is not a relevant element of the meteorological conditions in the Po valley. In addition, other meteorological variables have been introduced in the analysis: the diurnal temperature range, day-to-day mean temperature and mean pressure changes. More specifically, the diurnal temperature range corresponds to the difference between maximum and minimum temperature in each day. The day-to-day mean changes correspond to the difference between temperature or pressure values between each day and the previous one. As for air quality variables, particulate matters (PM10 and PM2.5), O_3_, nitrogen oxides (NOx), carbon monoxide (CO) and sulfur dioxide (SO_2_) have been selected in the statistical model being indicators of the pollution level. CO and nitrogen monoxide (NO) are primary pollutants that are most correlated with traffic fluxes. Particulate matters and NO_2_ are very important pollutants for the characterization of air quality in the Po valley area [12]. All these pollutants show a very pronounced seasonal effect, with the highest values during winter and smaller values during summer. On the other hand, O_3_ is the most relevant pollutant during summer, while is generally irrelevant during the remaining seasons. Table 1 contains a descriptive analysis of the meteorological and air quality conditions characterizing the different centers.

### 2.3. Statistical Analyses

Hospital admissions for VOC and ACS were correlated with daily meteorological and air quality data using a meta-analytical approach, which represents a standard, well-established statistical procedure to combine the results of independent studies referring to the same research hypothesis. Statistical analysis was performed using a quasi-likelihood Poisson regression in a generalized linear model to fit the effect of environmental parameters on hospital admissions. In addition, a distributed lag non-linear model (DLNM) was incorporated to explore nonlinear relationships and delayed effects. Specific libraries of R statistical software were used for the data analysis [17]. The effect of each environmental variable on the hospitalization risk was assessed from the day before the ED access (lag 1) back to 10 days in advance of the crisis (lag 10). The day of the hospitalization (lag 0) was not inserted in the analysis to avoid considering data recorded after the hospitalization event. Any single environmental variable was inserted in the DLNM individually to isolate each effect and its statistical significance when present. The assessment of the RR of ED admission was investigated through the evaluation at both extremes of the above-mentioned variables (comparison between 5° and 10° percentile and between 95° and 90° percentile). The increased risk is shown in the tables in the Results section to give emphasis to the tails of the statistical distributions and to show the effect of highest and lowest values on the increased risk of hospital admission. The values were considered statistically significant at 95% confidence level (CI95). The flu epidemic was considered as the only confounding factor in the statistical analysis.

## 3. Results

Clinical data included 615 ED accesses for VOC and/or ACS of 198 patients with SCD during a 5-year period of follow up (1/1/2011–31/12/2015, 1826 days in total). Patients were 155 HbSS patients, 15 HbSβ and 28 HbSC pediatrics and young adults (age range 0.18–31.9 years). The results of the meta-analysis are shown in Table 2 and Table 3.

Air quality variables seemed to have a lower impact with respect to meteorological variables: in particular, only O_3_ showed an increased risk of hospitalization at lag 5 both for high values with respect to average (+17%, CI95: 1.00–1.37) and for decreasing values at the tail, corresponding to the lowest concentration of this pollutant (+4%, CI95: 1.00–1.08). Since O_3_ has a well-defined seasonal trend, with the lowest values during winter and the highest during summer, it may be argued that this is the reason for the statistically significant effect on SCD. As for the impact of the selected meteorological variables on the hospital admission RR increase, there were several results statistically significant. First, a significant increase in the risk was observed at the lag closest to the hospital admissions for low maximum temperature (lag 3) and low minimum relative humidity (lag 1–2) with respect to average values: in both cases, a relevant increase in RR was present (Table 2). Additionally, for higher-than-average values of the mean wind speed, an increased RR was present at lag 1 (+19%, CI95: 1.01–1.40), but this was the only significant signal relative to the wind. Finally, the diurnal temperature range appeared as an important cause of hospital admission, with a statistically significant signal at lag 2 both for lower-than-average values (+17%, CI95: 1.00–1.36) and for higher-than-average values (+23%, CI95: 1.02–1.48), indicating that sudden temperature changes (positive and negative) were an important cause of hospital admissions (Table 2). Examining in more detail the tails of the variable distributions, other signals emerged. In particular, at lag 2 and lag 4 for the lowest values of the temperature difference from day to day (Figure 1), there was a small, significant increase in RR: a marked decrease in this variable in its lowest values caused a 6% and 5% increase in the RR at lag 2 (CI95: 1.00–1.12) and at lag 4 (CI95: 1.00–1.10).

No significant correlation was found for minimum temperature; only an increase in the highest values of mean and maximum temperatures was associated with an increase in RR value at lag 8 (+4%, CI95: 1–1.09) for maximum temperature and for mean temperature (+5%, CI95 :1–1.10), which was anyway quite far from the hospital admission event. Finally, from Table 3, low pressure values at lag 4 were associated with an increase in the RR of hospital admission.

An increase of 4% (CI95: 1–1.09) and 5% (CI95: 1.00–1.10) was found for average and minimum atmospheric pressure, respectively (see Figure 2 for average pressure).

The atmospheric pressure may be considered a proxy of bad weather conditions, and in this respect, a worsening of weather condition few days in advance can be the cause of hospital admission.

## 4. Discussion

The present study is the first attempt to quantifying the environmental impact on SCD in Italy. Few studies in the literature focused on the definition of the environmental effect in causing the onset of SCD crisis. Pioneering studies date back to the 70s–80s, but probably the small number of patients and the difficulties in gathering robust environmental data led to quite contrasting results [18,19,20]. More recently, the study of the effects of meteorological and air quality conditions benefited from conventional statistical analysis based on correlation and univariate and multivariate regression as well as of cutting-edge statistical techniques, such as generalized linear models to fit the effect of environmental parameters on hospital admissions using also distributed lag non-linear models to explore nonlinear relationships and delayed effects. Such types of statistical analysis rely on the availability of large database of patient’s hospitalization and also take advantage of the widespread diffusion of certified environmental data (especially for air quality). In our study, we considered several independent variables, taking into consideration that the database of ED access is not very large, and our goal was to screen only the significant effects of the environmental variables. The results obtained in this study showed some interesting aspects, partly confirming results from the literature, partly contrasting. Recently, a study suggested that the role of meteorological factors is more important than air pollutants [12]. This was confirmed also from our results, where only O_3_ showed a significant impact but only with a marked delay. In this respect, we specify that air quality conditions in the centers taking part to the study are quite similar: all are located in the Po Valley, one of the most polluted area in Europe [14]. Other studies failed to point out an association between the risk of hospitalization and a specific air pollutant with contrasting results: in a Paris study [10], the risk was significantly associated with high values of NO_2_, coarse (PM10) and fine (PM2.5) particulate matters and low values of CO, O_3_ and SO_2_, while in a London study, high levels of O_3_ and low levels of NO and CO showed the same effect [21]. Piel et al. mentioned only CO as having a significant effect on the risk of hospitalization [12]. O_3_ levels are higher during the summer season, worsening exacerbations of asthma, which can promote sickle cell crises. CO and NO inhibit vasoconstriction and platelet aggregation, thus having a potential protective role [22]. A recent large study conducted in Atlanta found a positive correlation between traffic related air pollution (CO, elemental carbon and NO_2_ levels) and SCD accesses, more significant in a pediatric population [13]. These contrasting results suggest that probably not all the aspects related to air quality factors are clear. Another important point is related to the effect of cold weather conditions, which show a more relevant effect with respect to heat. Several studies reported the association between crisis and exposure to cold conditions in various areas [9,10,11]. High wind speed was in turn considered very relevant in determining SCD crisis [3,9,10,23,24]. A weak association with wind was found in the present Italian study: this may be explained by the fact that relevant wind speed in the Po Valley is unusual especially in autumn and winter. On the other hand, cold temperatures were rather important (particularly, for low maximum temperature values). In our study, a large temperature drop from day to day and daily diurnal temperature range were associated with an increase of the hospitalization risk; these findings were reported also from a large study conducted in Paris describing that the sudden change in temperature from day to day and the presence of thunderstorms correlated with ED accesses [10]. As for humidity, other authors found a significant association with low values of this variable, although one may argue that no significant relationship with temperature or with atmospheric pressure emerged in this study [24]. Wind speed and low humidity are independent factors and cause skin cooling, vasoconstriction and blood flow deceleration of small vessels causing sickle cell crises [24]. The role of wind speed and rainfall was also confirmed by another large study conducted in London and Paris, whereas weak or no associations were found with temperatures values [12]. Most studies showed no significant association of pressure with hospitalization risk [12,24]. Nevertheless, it seems to be worth mentioning that the increase in hospitalization risk that emerged in this study is associated with low pressure values, which is a situation generally associated with bad weather conditions. Some limits have to be considered, such as the reduced sample size, the limited duration of the period considered and the retrospective analyses. Air quality parameters are mainly representative of local conditions due to the characteristics of the area where the monitoring stations are located. Traffic, or residential or background stations generally measure different pollutants and give rather different pollution levels for the pollutants monitored at different locations. The availability of air quality stations is crucial for these kinds of studies, which rely on the correct evaluation of people’s exposure to the various pollutants. Therefore, the choice of air quality stations may lead to relevant misclassification errors in attributing exposure to patients during their everyday life. This may also be a possible explanation for the sometimes-contrasting results obtained in the literature. The analyses considered only hospital access for patients with SCD, not considering acute clinical complications managed at home or with the support of the telephonic consultation from the physician. The small sample of patients included has limited the possibility to stratify the population based on genotype, presence of chronic therapy and clinical severity to study a possible different influence of environment. Indoor environmental conditions, tobacco smoke exposition and patients’ mobility are quite impossible to quantify but are potential influencing factors of clinical acute complications of these patients.

## 5. Conclusions

This study evidenced the role of low temperatures, low pressure, temperature excursions, low humidity values and high ozone levels in influencing the clinical course of SCD. Environmental factors are important determinants of the acute complications of SCD; however, the mode of action is still partially understood. Patients’ education and a better knowledge of the modes of actions of these factors could reduce hospital accesses and, therefore, contribute to the development of new therapies.

## Figures and Tables

**Figure 1 children-09-01478-f001:**
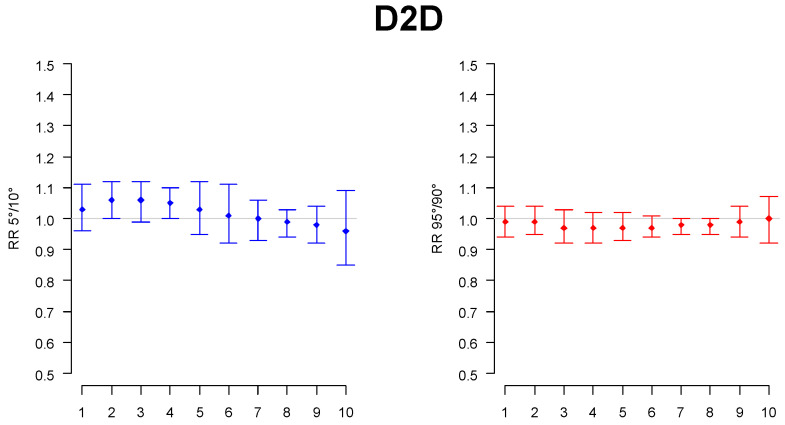
Relative risk of hospital admission in the lowest (“RR5vs10”, in blue) and highest tail of the variable distribution (“RR95vs90”, in red) for the temperature difference with respect to the previous day (D2D).

**Figure 2 children-09-01478-f002:**
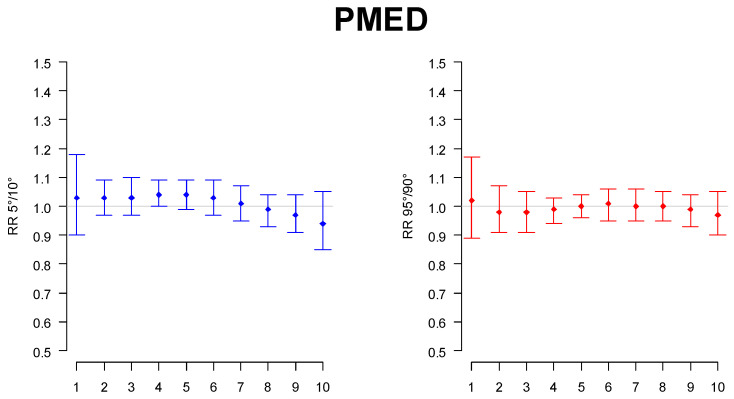
Relative risk of hospital admission in the lowest (“RR5vs10”, in blue) and highest tail of the variable distribution (“RR95vs90”, in red) for average atmospheric pressure.

**Table 1 children-09-01478-t001:** Mean Value (“*M*”), standard deviation (“*SD*”), minimum (“*min*”) and maximum (“*max*”) of the distribution of meteorological and air quality variables considered in the meta-analysis as predictor of the hospitalization risk for SCD in the different centers. “N/A” indicates that the corresponding values are not measured or available in the center. DTR: diurnal temperature range, D2D: day-to-day mean temperature changes, DPRESS day-to-day mean pressure changes, PM10: particulate matters with a diameter of 10 μm or less, PM2.5 particulate matters with a diameter of 2.5 μm or less. O_3_: ozone. NO: Nitrogen oxide, NOx: Nitrogen oxides, NO_2_: nitrogen dioxide, CO: carbon monoxide, SO_2_: sulphur dioxide.

	*Modena/Parma*				*Monza*				*Padova*			
	*M*	*SD*	*min*	*max*	*M*	*SD*	*min*	*max*	*M*	*SD*	*min*	*max*
**PRECIPITATION**	1.7	5.0	0	48.4	3.3	9.0	0	109.0	n/a	n/a	n/a	n/a
**PRESSURE**												
**Max**	1010.2	7.0	977.2	1030.3	1010.1	7.1	977.0	1030.7	1018.5	7.1	987.4	1039.1
**Med**	1007.7	7.3	971.0	1028.7	1007.6	7.3	971.5	1028.5	1016.4	7.4	979.2	1037.6
**Min**	1005.2	7.8	967.6	1027.6	1005.1	7.7	968.3	1027.5	1014.4	7.9	975.3	1036.8
**TEMPERATURE**												
**Max**	19.6	9.5	−3.7	39.3	18.8	8.7	−1.7	36.8	18.3	9.1	−3.0	38.2
**Med**	15.0	8.4	−6.3	31.9	14.1	7.9	−6.0	31.1	14.0	7.9	−4.8	31.5
**Min**	10.8	7.4	−9.5	26.0	10.0	7.2	−9.4	25.9	10.2	7.0	−8.1	25.8
**HUMIDITY (U)**												
**Max**	81.6	13.3	26.0	98.0	86.8	12.7	22.1	100.0	91.2	12.0	29.0	100.0
**Med**	63.5	18.2	16.5	97.0	68.8	17.4	15.9	100.0	71.6	14.6	22.5	100.0
**Min**	44.3	21.3	6.0	97.0	48.8	20.1	9.3	100.0	52.0	20.6	12.0	100.0
**WIND SPEED (V)**												
**Max**	8.2	3.2	2.8	24.5	2.0	0.8	0.2	5.6	7.8	3.4	2.3	32.7
**Med**	2.2	0.8	0.6	8.7	0.9	0.5	0.1	4.1	2.3	1.0	0.2	10.3
**Min**	0	0	0	0.8	0.1	0.2	0	2.5	n/a	n/a	n/a	n/a
**DTR**	8.9	3.5	0.8	16.2	8.9	3.5	0.5	18.0	8.2	3.2	0.9	16.4
**D2D**	0	1.8	−12.0	6.3	0	1.7	−10.0	8.2	0	1.8	−11.3	6.7
**DPRESS**	0	4.3	−20.0	17.9	0	4.3	−19.1	18.8	0	4.3	−22.0	18.1
**PM_10_**	30.8	19.0	2.0	168.0	39.8	27.8	3.0	199.0	37.9	25.4	2.0	175.0
**PM_2.5_**	21.0	15.7	1.0	133.0	31.5	23.7	0	199.0	29.6	23.6	4.0	193.0
**O_3_**	43.5	29.0	2.0	124.0	42.8	33.3	1.9	156.6	51.6	27.8	2.0	132.0
**NO**	22.3	28.4	0	236.0	n/a	n/a	n/a	n/a	22.9	33.8	1.0	251.0
**NO_X_**	64.2	56.0	6.0	453.0	99.8	99.6	10.8	707.5	69.5	62.8	11.0	476.0
**NO_2_**	30.1	15.3	2.0	103.0	46.2	22.9	10.5	169.1	34.7	14.2	10.0	91.0
**CO**	0.5	0.3	0.1	1.8	0.8	0.5	0.1	4.7	0.6	0.3	0.1	2.5
**SO_2_**	5.4	2.8	0	23.0	5.2	2.2	0	15.2	2.2	0.7	2.0	9.0

**Table 2 children-09-01478-t002:** Relative risk (RR) of hospital admission for low variable values (“RR10vs50”) and for high variable values (“RR90vs50”) with respect to average. A RR value greater than 1 indicates that the risk of hospitalization is increasing when moving towards lower (“RR10vs50“) or higher (“RR90vs50”) values with respect to the risk associated with average (i.e., median) values. UMIN: minimum relative humidity; TMAX: maximum temperature; VMED: mean wind speed; DTR: diurnal temperature range; O_3_: ozone. CI_inf is the lower boundary of the 95% confidence level interval, while CI_sup is the upper boundary of the same interval; the intervals indicating a statistically significant increase of RR are highlighted.

UMIN	**Lag**	**RR10vs50**	**CI_inf**	**CI_sup**	**RR90vs50**	**CI_inf**	**CI_sup**
	1	1,34	1,07	1,66	0,99	0,72	1,37
	2	1,20	1,07	1,34	1,19	0,98	1,44
	3	1,07	0,9	1,28	1,19	0,93	1,54
	4	1,05	0,94	1,17	1,06	0,89	1,27
	5	1,05	0,97	1,14	0,97	0,84	1,11
	6	1,04	0,96	1,14	0,92	0,79	1,06
	7	1,03	0,95	1,12	0,91	0,79	1,04
	8	1,00	0,94	1,08	0,92	0,79	1,07
	9	0,98	0,88	1,08	0,95	0,77	1,16
	10	0,95	0,81	1,12	1,01	0,77	1,32
TMAX	**Lag**	**RR10vs50**	**CI_inf**	**CI_sup**	**RR90vs50**	**CI_inf**	**CI_sup**
	1	0,55	0,30	1,03	1,06	0,64	1,77
	2	1,06	0,83	1,36	1,01	0,80	1,29
	3	1,43	1,03	1,99	1,04	0,75	1,44
	4	1,22	0,90	1,65	1,08	0,90	1,31
	5	0,99	0,78	1,24	1,12	0,94	1,33
	6	0,86	0,69	1,07	1,13	0,92	1,38
	7	0,80	0,66	0,97	1,10	0,91	1,32
	8	0,79	0,68	0,92	1,05	0,90	1,21
	9	0,81	0,67	0,98	0,98	0,80	1,19
	10	0,84	0,61	1,16	0,91	0,66	1,25
VMED	**Lag**	**RR10vs50**	**CI_inf**	**CI_sup**	**RR90vs50**	**CI_inf**	**CI_sup**
	1	0,87	0,73	1,03	1,19	1,01	1,40
	2	0,89	0,81	0,99	1,02	0,92	1,14
	3	0,94	0,84	1,05	0,95	0,85	1,08
	4	0,98	0,85	1,12	0,97	0,88	1,08
	5	1,00	0,89	1,11	0,99	0,90	1,08
	6	1,00	0,90	1,11	0,99	0,89	1,10
	7	0,98	0,90	1,08	0,98	0,89	1,09
	8	0,96	0,88	1,04	0,96	0,87	1,05
	9	0,92	0,82	1,04	0,92	0,83	1,02
	10	0,89	0,76	1,04	0,88	0,76	1,02
DTR	**Lag**	**RR10vs50**	**CI_inf**	**CI_sup**	**RR90vs50**	**CI_inf**	**CI_sup**
	1	0,89	0,66	1,20	1,27	0,95	1,70
	2	1,17	1,00	1,36	1,23	1,02	1,48
	3	1,22	0,96	1,56	1,11	0,93	1,33
	4	1,08	0,85	1,36	1,05	0,91	1,22
	5	0,95	0,77	1,17	1,03	0,90	1,18
	6	0,89	0,74	1,08	1,02	0,88	1,18
	7	0,89	0,76	1,04	1,02	0,88	1,17
	8	0,92	0,80	1,06	1,03	0,90	1,17
	9	0,98	0,84	1,14	1,05	0,92	1,21
	10	1,06	0,87	1,30	1,08	0,89	1,31
O_3_	**Lag**	**RR10vs50**	**CI_inf**	**CI_sup**	**RR90vs50**	**CI_inf**	**CI_sup**
	1	0,69	0,42	1,13	1,48	0,96	2,28
	2	0,84	0,66	1,06	1,25	0,90	1,73
	3	1,02	0,74	1,41	1,08	0,83	1,40
	4	1,09	0,87	1,36	1,11	0,94	1,32
	5	1,13	0,84	1,51	1,17	1,00	1,37
	6	1,11	0,78	1,58	1,17	0,97	1,42
	7	1,07	0,75	1,52	1,15	0,95	1,38
	8	1,01	0,75	1,36	1,10	0,92	1,30
	9	0,93	0,73	1,18	1,02	0,83	1,26
	10	0,86	0,64	1,16	0,95	0,69	1,29

**Table 3 children-09-01478-t003:** Relative risk of hospital admission in the lowest (“RR5vs10”) and highest tail of the variable distribution (“RR95vs90”). PMED: mean atmospheric pressure; PMIN: minimum atmospheric pressure, TMAX maximum temperature; TMED: mean temperature; D2D: temperature difference with respect to the previous day; O_3_: ozone. ICinf: inferior extreme of the 95% level confidence interval; ICsup: superior extreme of the 95% level confidence interval; intervals above 1 are highlighted.

PMED	**Lag**	**RR5vs10**	**CI_inf**	**CI_sup**	**RR95vs90**	**CI_inf**	**CI_sup**
	1	1,03	0,90	1,18	1,02	0,89	1,17
	2	1,03	0,97	1,09	0,98	0,91	1,07
	3	1,03	0,97	1,10	0,98	0,91	1,05
	4	1,04	1,00	1,09	0,99	0,94	1,03
	5	1,04	0,99	1,09	1,00	0,96	1,04
	6	1,03	0,97	1,09	1,01	0,95	1,06
	7	1,01	0,95	1,07	1,00	0,95	1,06
	8	0,99	0,93	1,04	1,00	0,95	1,05
	9	0,97	0,91	1,04	0,99	0,93	1,04
	10	0,94	0,85	1,05	0,97	0,90	1,05
PMIN	**Lag**	**RR10vs50**	**CI_inf**	**CI_sup**	**RR90vs50**	**CI_inf**	**CI_sup**
	1	1,01	0,89	1,14	1,04	0,91	1,19
	2	1,06	0,99	1,12	1,00	0,92	1,08
	3	1,06	0,99	1,13	0,98	0,91	1,05
	4	1,05	1,00	1,10	0,98	0,94	1,03
	5	1,03	0,97	1,09	0,99	0,95	1,03
	6	1,01	0,94	1,08	1,00	0,95	1,05
	7	0,99	0,92	1,06	1,00	0,95	1,05
	8	0,97	0,91	1,04	1,00	0,96	1,05
	9	0,96	0,90	1,03	0,99	0,94	1,05
	10	0,94	0,85	1,04	0,99	0,92	1,07
TMAX	**Lag**	**RR10vs50**	**CI_inf**	**CI_sup**	**RR90vs50**	**CI_inf**	**CI_sup**
	1	0,99	0,84	1,15	1,11	0,82	1,49
	2	1,04	0,95	1,15	1,02	0,91	1,16
	3	1,01	0,88	1,16	0,98	0,83	1,17
	4	0,97	0,88	1,07	0,99	0,89	1,10
	5	0,95	0,87	1,03	1,01	0,95	1,07
	6	0,95	0,87	1,03	1,02	0,95	1,10
	7	0,97	0,92	1,02	1,04	0,97	1,11
	8	0,99	0,95	1,03	1,04	1,00	1,09
	9	1,03	0,96	1,10	1,05	0,99	1,11
	10	1,07	0,94	1,23	1,06	0,94	1,19
TMED	**Lag**	**RR10vs50**	**CI_inf**	**CI_sup**	**RR90vs50**	**CI_inf**	**CI_sup**
	1	0,94	0,77	1,15	1,12	0,86	1,44
	2	1,03	0,94	1,12	1,04	0,90	1,20
	3	1,04	0,95	1,14	0,97	0,86	1,10
	4	1,01	0,95	1,06	0,99	0,91	1,08
	5	0,96	0,89	1,03	1,01	0,94	1,09
	6	0,95	0,89	1,02	1,03	0,93	1,13
	7	0,96	0,92	1,02	1,04	0,96	1,13
	8	0,99	0,95	1,03	1,05	1,00	1,10
	9	1,01	0,94	1,09	1,05	0,98	1,13
	10	1,06	0,94	1,21	1,06	0,90	1,23
D2D	**Lag**	**RR10vs50**	**CI_inf**	**CI_sup**	**RR90vs50**	**CI_inf**	**CI_sup**
	1	1,03	0,96	1,11	0,99	0,94	1,04
	2	1,06	1,00	1,12	0,99	0,95	1,04
	3	1,06	0,99	1,12	0,97	0,92	1,03
	4	1,05	1,00	1,10	0,97	0,92	1,02
	5	1,03	0,95	1,12	0,97	0,93	1,02
	6	1,01	0,92	1,11	0,97	0,94	1,01
	7	1,00	0,93	1,06	0,98	0,95	1,00
	8	0,99	0,94	1,03	0,98	0,95	1,00
	9	0,98	0,92	1,04	0,99	0,94	1,04
	10	0,96	0,85	1,09	1,00	0,92	1,07
O_3_	**Lag**	**RR10vs50**	**CI_inf**	**CI_sup**	**RR90vs50**	**CI_inf**	**CI_sup**
	1	0,96	0,88	1,06	1,03	0,78	1,35
	2	0,99	0,94	1,04	1,03	0,90	1,18
	3	1,02	0,96	1,08	1,00	0,86	1,16
	4	1,03	0,99	1,07	0,99	0,89	1,11
	5	1,04	1,00	1,08	0,99	0,91	1,08
	6	1,03	0,98	1,08	1,01	0,90	1,12
	7	1,02	0,97	1,07	1,02	0,90	1,16
	8	1,00	0,96	1,04	1,03	0,91	1,18
	9	0,98	0,94	1,02	1,04	0,91	1,19
	10	0,95	0,89	1,02	1,06	0,90	1,25

## Data Availability

Not applicable.

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
