# Peer review of "Environmental Factors in Northern Italy and Sickle Cell Disease Acute Complications: A Multicentric Study"

_children, 2022, doi:10.3390/children9101478_

Round 1

Reviewer 1 Report

This is an interesting and well presented study.   My only comment, is that when it comes to air pollution factors, analysis of only a few days prior to acute presentations may not suffice to show any significant differences.   Long-term exposure to different pollutants(if such differences exist in the examined regions) and the long-term rate of presentations may be more informative. 

Author Response

Reviewer “1”: This is an interesting and well presented study. My only comment is that when it comes to air pollution factors, analysis of only a few days prior to acute presentations may not suffice to show any significant differences. Long-term exposure to different pollutants (if such differences exist in the examined regions) and the long-term rate of presentations may be more informative. 

We sincerely thank you for the appreciation of the strength of our study.

Your comment is an interesting point and probably long-term exposure needs further investigation. Nevertheless, the application of more ad hoc statistical techniques should be the correct approach. In fact, other published papers having methodological frameworks similar to us, apply statistical models to show lagged effects only for 14 days in [8], 7 days in [10] and 3 days in [11] since the focus of the analysis is the acute (short-term) effects, like in our study.

Reviewer 2 Report

In this multicentric study, authors have targeted four Pediatric Onco-Hematology Centers located in the cities of Modena, Parma, Monza and Padova of Northern Italy for evaluating influence of environment factor such as atmospheric pressure, temperature, relative humidity and wind velocity within young adult patients. The results obtained in this study showed some interesting aspects. This study evidenced the role of low temperatures, low pressure, temperature excursions, low humidity values and high ozone levels in influencing the clinical course of SCD. However, some suggestions must be incorporated by the authors to improve the manuscript.

Comments:

1.     I advise authors to mention p value for each analysis in result and discussion section

2.     I recommend authors to mention the prevalence of SCD in the introduction section with proper reference citation.

3.     I suggest authors to mention the calculation through which they determined composite index of temperature, wind chill index and relative humidity within the manuscript.

4.     What are the air quality factors addressed by the authors in the study?

5.     Did authors evaluate air quality factors like daily mean concentrations of carbon monoxide, nitrogen oxides, sulfur dioxide, and ozone particle matter and black carbon in this study?

6.     Authors can cite few recent review articles on Sickle cell disease and its pathophysiological aspects. [e.g. PMID: 32157419, PMID: 24252885]

Author Response

Reviewer “2”: In this multicentric study, authors have targeted four Pediatric Onco-Hematology Centers located in the cities of Modena, Parma, Monza and Padova of Northern Italy for evaluating influence of environment factor such as atmospheric pressure, temperature, relative humidity and wind velocity within young adult patients. The results obtained in this study showed some interesting aspects. This study evidenced the role of low temperatures, low pressure, temperature excursions, low humidity values and high ozone levels in influencing the clinical course of SCD. However, some suggestions must be incorporated by the authors to improve the manuscript.

We are really grateful for your positive comments on our paper. Taking into account your comments, we revised the paper

I advise authors to mention p value for each analysis in result and discussion section Answer:  A single p-level for statistical significance using dlnm approach is not available (as it would be the case in a t-test or in a correlation test). In this approach, the confidence interval at 95% level (CI95) for the relative risk (RR) is the usual indicator of statistical significance (examples are at lines 138, 139, 148, 151, 157, 163, 164 and 172 in the original manuscript). This corresponds to state the specific RR values are statistically significant at 95% level whenever its value is inside the reported CI95 interval. We modified Table 2 and 3 in order to be fully consistent with the manuscript.

I recommend authors to mention the prevalence of SCD in the introduction section with proper reference citation. Answer: As you suggested, we added a more detailed sentence in the text.

I suggest authors to mention the calculation through which they determined composite index of temperature, wind chill index and relative humidity within the manuscriptAnswer: We added a more detailed sentence in the text starting at line 86 (original manuscript).

What are the air quality factors addressed by the authors in the study? Answer: The list of air quality factors introduced in the analysis is in Sect 2.2., lines 86-92 (original manuscript). We considered only the daily concentration of these pollutants in the dlnm model.

Did authors evaluate air quality factors like daily mean concentrations of carbon monoxide, nitrogen oxides, sulphur dioxide, and ozone particle matter and black carbon in this study? Answer: We tested particulate matters, namely PM10 and PM2.5, ozone O3, nitrogen oxides NOX, carbon monoxide CO and sulphur dioxide SO2 in the dlnm model, but we decided to stick to statistical significant results only. We obtained no statistically significant results for air quality parameters except for O3, as we stated. As for Black Carbon, routine daily measures are not available in Italy, except for specific monitoring campaign and for limited periods. Unfortunately, it was not the case for the years considered.

Authors can cite few recent review articles on Sickle cell disease and its pathophysiological aspects. [e.g. PMID: 32157419, PMID: 24252885]: Answer: We added the reference

Round 2

Reviewer 2 Report

Authors have revised the manuscript considering the comments and suggestions given.